# Influences of Electron Beam Irradiation on the Physical and Chemical Properties of Zearalenone- and Ochratoxin A-Contaminated Corn and In Vivo Toxicity Assessment

**DOI:** 10.3390/foods9030376

**Published:** 2020-03-24

**Authors:** Xiaohu Luo, Yuheng Zhai, Lijun Qi, Lihong Pan, Jing Wang, Jiali Xing, Ren Wang, Li Wang, Qingchuan Zhang, Kai Yang, Zhengxing Chen

**Affiliations:** 1Beijing Advanced Innovation Center for Food Nutrition and Human Health, Beijing Technology and Business University (BTBU), Beijing 100048, China; wangjing@th.btbu.edu.cn; 2National Engineering Laboratory for Cereal Fermentation Technology, Jiangnan University, Wuxi 214122, China; 6190112167@stu.jiangnan.edu.cn (Y.Z.); ljqi0510@sohu.com (L.Q.); yelanplh@outlook.com (L.P.); nedved_wr@jiangnan.edu.cn (R.W.); legend0318@hotmail.com (L.W.); yangkai164@outlook.com (K.Y.); zxchen@jiangnan.edu.cn (Z.C.); 3National Engineering Laboratory for Agri-product Quality Traceability, Beijing Technology and Business University, Beijing 100048, China; zqc1982@126.com; 4Ningbo Institute for food control, Ningbo 315048, China; hellojiali77@gmail.com

**Keywords:** zearalenone, ochratoxin A, electron beam irradiation, physical and chemical properties, toxicity in vivo

## Abstract

Electron beam irradiation (EBI) has high energy, no induced radioactivity, and strong degradation capacity toward mycotoxins, such as zearalenone (ZEN) and ochratoxin A (OTA). In this study, we determined EBI’s influence on the physical and chemical properties of corn contaminated with ZEN and OTA. Moreover, the toxicity of corn after EBI was assessed through a mouse experiment. Amylose content and starch crystallinity in corn decreased significantly (*p* < 0.05) at an irradiation dose higher than 20 kGy. Scanning electron microscopy results revealed that the starch particles of corn began to be crushed at 10 kGy. Essential and total amino acid contents in corn decreased significantly with increasing irradiation dose of EBI (*p* < 0.05). Feeding EBI-treated corn fodders to mice could significantly improve blood biochemical indexes. The EBI-treated group was not significantly different from the normal corn group and did not display histopathological changes of the liver. EBI treatment can influence the quality of corn to some extent and effectively lower the toxicity of ZEN and OTA in contaminated corn. The results provide a theoretical and practical basis for the processing of EBI-treated corn and its safety.

## 1. Introduction

Pollution of bulk foods by mycotoxin has become a global concern and causes considerable economic loss every year. As mycotoxins with strong toxicity, zearalenone (ZEN) and ochratoxin A (OTA) have extremely extensive distributions [1]. ZEN is a secondary metabolism product produced by *Fusarium* strains and can be detected in cereals like corn, barley, and wheat. Corn is predominantly contaminated by ZEN, and most regions in China are exposed to ZEN pollution [2]. OTA is produced by *Aspergillus* and *Penicillium* fungi during storage of cereals and associated products [3,4,5]. Food and food products are substantially contaminated by OTA [6]. ZEN can cause many symptoms (e.g., nausea, emesis, and diarrhea) in humans and animals, abnormal reproductive functions (e.g., abortions and dead fetuses) of animals, poor growth, immunity inhibition, sterility, and malformation [7,8]. OTA is known to be hepatotoxic, nephrotoxic, teratogenic, mutagenic, and carcinogenic [9].

ZEN and OTA readily pollute most cereal crops with high global yield, such as corn [10]. Scholars have investigated the pollution of 7.049 mycotoxins in corn, soybean meal, and wheat fodders from 2009 to 2011. The detection rates of ZEN and OTA were 45% and 28%, respectively [11]. Researchers [12] who investigated mycotoxin pollution in South Africa in one decade (2007−2016) confirmed that *Fusarium* and *Aspergillus* fungi are the main pollutants of corn. In Zimbabwe, the ZEN detection rate in corn samples was 15%, and the average ZEN concentration was 110 μg/kg. An investigation revealed that OTA exists from a cold temperate zone to a tropical zone (North America and South America, North Europe and West Europe, and Africa and South Asia) and contaminates cereals and relevant products [13]. Considering the serious worldwide pollution of foods and food products by mycotoxins, most countries have proposed strict regulations on mycotoxin limits. In China, the ZEN limit in cereal and relevant products is 60 μg/kg, and the OTA limit in cereal and relevant grinding products is 5 μg/kg [14].

Currently, ZEN and OTA pollution is difficult to control by regional and weather conditions. Moreover, ZEN and OTA have stable properties and are difficult to degrade, thereby posing considerable threats to human and animal health and economic development. Electron beam irradiation (EBI) has been applied in the degradation of mycotoxins in recent years. EBI can adequately maintain the freshness and edible quality of foods without pollution and residues due to it’s the slight temperature rise it causes compared with γ-irradiation [15]. In addition, no induced radioactivity is involved in the safe dose. EBI is sanitary and safe. Xue et al. studied the influences of 0−5.40 kGy EBI treatment on the gelatinization and the physical and chemical properties of corn flour. They found that the total content of starch and crude fibers in corn flour declined dramatically (*p* < 0.05), whereas the moisture and reducing sugar contents increased significantly (*p* < 0.05) [16]. Thus, EBI treatment can influence the physical and chemical properties of corn. According to previous studies, the mice model can be used for the in vivo toxicity assessment of ZEN [17,18].

Many studies concerning the influences of EBI treatment on ZEN- and OTA-contaminated corn have been conducted [19]. However, the influences of EBI treatment on the physical and chemical properties of contaminated corns have not been extensively studied, and no relevant toxicity has been reported. Thus, the present work analyzed the physical and chemical properties of contaminated corn after EBI treatment. In addition, ZEN- and OTA-contaminated corns and contaminated corns after EBI treatment were fed to mice, and toxicity was subsequently assessed to explore the quality changes and safety of contaminated corns following EBI treatment. This study can provide theoretical reference and practical basis for the safety of corn and processed products processed by EBI.

## 2. Materials and Methods 

### 2.1. Materials and Reagents

ZEN- and OTA-contaminated corn kernels and non-contaminated corn kernels were collected from farmer’s markets in Taixing City, Jiangsu Province, China. ZEN and OTA standard samples (purity ≥ 99.8%) and acetic acid (chromatographically pure) were provided by Bailingwei Technology Co., Ltd. (Beijing, China). Methyl alcohol and acetonitrile (chromatographically pure) were produced by Fisher Scientific Company (Waltham, MA, USA). Ultrapure water (resistance ≥ 18.2 MΩ/cm) was prepared by using the Simplicity UV ultrapure water instrument (Millipore Company, Bedford, MA, USA). Nitrogen (purity ≥ 99.8%) was bought from Wuxi Shinnai Chemical Gas Co., Ltd, Wuxi, China. Other analytically pure reagents were bought from Sinopharm Group Chemical Reagent Co., Ltd.(Shanghai, China) unless otherwise specified.

Blood biochemical reagents were imported from WAKO Pure Chemical Industries, Ltd. (Kanagawa, Japan) with original packaging. ICR (Institute of Cancer Research) male mice (clean level) were bought from Shanghai Slyke Experimental Animal Co., Ltd., Shanghai, China. The license number of animal production is SCXK (Shanghai) 2012−0002, and the animal qualification certificate number is 20150000522095.

Basic and custom-made fodders were provided by Jiangsu Xietong Medicine Bioengineering Co., Ltd., Jiangsu, China. Custom-made fodders consist of 50% corn, 30% maintaining basic mice fodders, 2% salad oil, 10% casein, 2% crude fiber, 3% experimental animal gunk, and 3% calcium bicarbonate.

### 2.2. Main Instruments and Equipment

High-performance liquid chromatograph (HPLC) 1260 series with fluorescence detector (FLD) and ZORBAX SB C18 chromatographic column (150 mm × 4.6 mm; particle size: 5 μm) were manufactured by Agilent Company (Palo Alto, CA, USA). The EBI accelerator (AB5.0) was manufactured by Wuxi ELPONT Radiation Technological Co., Ltd. (Wuxi, China). X-ray diffractometer (D8) was manufactured by Bruker (China) AXS Co., Ltd., Beijing, China. High-resolution tungsten filament scanning electron microscope (SU3800) was manufactured by Techcomp (China) Scientific Instrument Co., Ltd., Tianjing, China.

### 2.3. EBI Treatment of Contaminated Corn

Corn kernels (200 g) with different moisture contents (13.9% and 19.1%) were placed in polyethylene bags and flattened to approximately 0.5 cm thickness. In addition, corn kernels were ground into powder and then sieved through a 30-mesh (0.6 mm) screen. An equal amount of the corn flour was placed in polyethylene bags and flattened before EBI treatment. The above samples were radiated under different doses (0, 5, 10, 20, 30, and 50 kGy). The accelerated electron energy, beam flow, and dose rates were set to 5 MeV, 20 mA, and 2 kGy/s, respectively. All radiated samples were tested at 4 °C.

### 2.4. Determination of OTA and ZEN in Corn and Fodders

ZEN content was tested by GB 5009.209–2016 Test of Zearalenone in Foods [20]. OTA content was tested by GB 5009.96–2016 Test of Ochratoxin A in Foods [21].

### 2.5. Test of Physical and Chemical Properties of Corn

#### 2.5.1. Amylose Content Test

Amylose content was tested by GB/T 15683–2008 Test of Amylose Content in Rice [22]. Starch content in corn was tested by GB/T 5009.9–2016 Test of Starch in Foods [23].

#### 2.5.2. Starch Crystallinity Test

Corn flour was kept under constant humidity for 48 h after grinding and sieving. Sheet samples of corn flour were made, and starch was scanned using an X-ray diffractometer within the range of 4°–40°.

#### 2.5.3. Observation of Starch Particles

The extracted corn starch was ground and sieved. After metal spraying, pictures were obtained using a scanning electron microscope. The pictures were magnified by 600 and 2400 times.

#### 2.5.4. Test of Amino Acids in Corn

Approximately 100 mg of corn flour was dissolved into a hydrolysis tube to which 8 mL of 6 mol/L HCL was added and nitrogen was supplied for 3 min. Then, the tube was sealed and hydrolyzed at 120 °C for 2 h. Next, 4.8 mL of 10 mol/L NaOH was added for neutralization, and the mixture was transferred to a 25 mL volumetric flask for constant volume. The supernatant was filtered, centrifuged, and finally transferred to a brown sample bottle for HPLC analysis.

### 2.6. Mouse Experiment

#### 2.6.1. Animal Grouping and Culture

In this experiment, 48 male ICR mice (clean level) were bought from Shanghai Slyke Experimental Animal Co., Ltd., China. They were randomly divided into six groups according to weight, and eight mice were assigned per group (Table 1).

All experimental animals were cultured with a barrier system in the Animal Experimental Center of Jiangnan University with the following conditions: indoor temperature of 22 ± 2 °C, relative humidity of 50%−60%, and day–night alternation hours of 12 h/12 h. The mice ate food and drank water freely and were fed with fodder daily. Every week, water was changed twice, and padding was changed once.

This study was approved by the Ethical Committee of Jiangnan University and the Laboratory Animal Management Committee of Jiangsu Province, China (Approval No. 2110748). All animal experiments were conducted in compliance with the standard ethical guidelines under the control of the aforementioned ethical committees.

#### 2.6.2. Serum and Routine Blood Biochemical Examination

The mice were fasted one night before the blood samples were collected from their eyeballs. The samples were placed in centrifuge tubes with an anticoagulant. The blood samples were centrifuged for 10 min at a rate of 3.000 r/min to obtain the surface blood plasma. The samples were reacted in a reaction cup under 37 °C. Various blood routine indexes were tested using an automatic biochemical analyzer.

#### 2.6.3. Organ Index and Histopathological Examination

The mice were sacrificed by breaking their necks. The livers, kidneys, spleens, and testes were collected and weighted. These organs were rinsed with normal saline and dried with a piece of filtering paper. Some liver and kidney tissues were cut and fixed in 4% neutral formaldehyde solution. The pathological changes of the tissues were observed under a TEM after HE dyeing.

### 2.7. Data Processing

All experimental procedures were conducted according to the principles of green analytical chemistry [24,25]. The results are expressed as mean ± standard deviation. Data were analyzed by the SPSS 16.0 software. The inter-group difference was analyzed by one-way analysis of variance. *p* ≤ 0.05 indicates significant difference.

## 3. Results and Discussion

### 3.1. Effects of EBI on the Physical and Chemical Properties of Corn

#### 3.1.1. Effects of EBI on Corn Amylose Content

The amylose content in corn decreased significantly with increasing irradiation dose (*p* < 0.05), as shown in Table 2. When EBI treatment doses were set to 10 and 20 kGy, the amylose content decreased significantly. With increasing irradiation dose, the reduction of amylose content decelerated. This outcome might be due to the tight distribution of some amylose inside the corn particles, thereby resulting in the poor action of irradiation rays. In 1991, Grant and D’appolonia studied the influences of low-level γ-irradiation on water-soluble non-starch polysaccharides separated from wheat flour and wheat membrane and confirmed that some particles break along the leaf molecules. Another study revealed that EBI may change the amylose content in rice because irradiation destroys the double helix structure of amylose, thereby reducing the amylose content [26]. Xue et al. found that under the influence of a high-power electron beam, starch particles were decomposed, and the amylose structure was destroyed [16]. Nemtanu et al. explored the degradation laws of solid amylose under accelerated EBI at the dose range of 10−50 kGy. Amylose structures were broken due to irradiation, and smaller pieces appeared [27]. These findings are consistent with the amylose content reduction in this study.

#### 3.1.2. Effects of EBI on Starch Crystallinity of Corn

Starch crystallinity decreased to some extent with increasing EBI dose (Table 3). This outcome might be due to the fact that starch crystallinity is related to amylopectin and that amylose is distributed in non-crystalline regions. EBI can destroy the long chain of amylopectin and decrease amylopectin content, thereby breaking the molecular chains of starch and its crystalline structure. As a result, crystallinity decreases. By using infrared spectroscopic analysis, Abu et al. found that the irradiation dose did not affect the degree of surface order (crystallinity) of starch particles [28]. Chung et al. processed potato and bean starches by using 50 kGy γ-ray irradiations. The relative crystallinity, degree of the order of particle surface, and gelatinization enthalpy decreased with increasing irradiation dose [29]. The different variation laws of crystallinity in the abovementioned studies might be due to the variation in the starch types. Starch crystallinity is negatively correlated with amylose content. However, the amylose content in corn decreased after EBI treatment. The double helix structures of amylose were destroyed after irradiation, as manifested by the reduced amylose content.

#### 3.1.3. Effects of EBI on the Appearance of Corn Starch Particles

The SEM of corn starch is shown in Figure 1. Figure 1A shows the corn starch particles. Figure 1B–D show the SEM images of samples under 10, 30, and 50 kGy, respectively. In each group of images, starch particles are magnified by 600 times (left-hand images) and 2400 times (right-hand images). Typical corn starch particles are polyhedrons with smooth corners, and some immature particles are spherical (Figure 1). The particle surface is slightly uneven due to small pits and small holes at the center of the particles. Corn starch particles after EBI treatment are broken. Starch appearance is destroyed with increasing irradiation dose. Hu et al. found that some starch particle surfaces cracked and displayed rough surfaces when the irradiation dose reached 4.4 kGy [30]. Sokhey et al. discovered from SEM that the γ-irradiation dose of 31 kGy did not destroy the physical structure of waxy corn starch particles, and damages to starch particles under irradiation altered the molecular structure of starch [31]. Chung et al. conducted a further study and confirmed that waxy corn was substantially influenced by γ-irradiation, thereby indicating that amylopectin was the starch component that was predominantly affected by irradiation. Amylopectin was broken down into short amylose after γ-irradiation [32]. Moreover, the morphology of starch particles remained unchanged after irradiation. This outcome might be due to the fact that irradiation energy reaches internal starch crystals. These results differed from those in the present study, and such variation might be related to the different proportions of amylose and amylopectin in corn. The corn samples used in the current research have lower amylopectin content than waxy corn, and irradiation energy destroys the apparent structure directly.

#### 3.1.4. Effects of EBI on Amino Acid Content

Changes of amino acid content in corn after EBI are shown in Table 4. EBI significantly influences amino acid in corn. With increasing irradiation dose, the essential and total amino acid contents in corn decreased significantly (*p* < 0.05), but their ratio changed slightly. Li Chao discovered that EBI could remarkably influence the essential amino acids of the human body and could induce the increase of linoleic, linolenic, and arachidonic acids in the dose range of 5−9 kGy. When the irradiation dose exceeded 9 kGy, the relative contents of the three essential fatty acids increased by 94.61%, 41.37%, and 89.91% [33]. Free radicals may be produced throughout the high-energy EBI. These free radicals could attack the side-chain groups of amino acid, occur in amidogen and the carboxyl groups of α-carbon, and induce amino acid deamination. Therefore, the amino acid content decreases [34]. In addition, irradiation under aerobic conditions may induce the oxidation of amino acids, especially aromatic, sulfur-containing, and aliphatic amino acids. Hooshmand et al. reported that the amino acid content in cereal changes substantially after irradiation. In particular, the Lys, Met, and Phe contents in corn decreased after irradiation at 7.5 kGy [35]. The abovementioned studies achieved generally the same results as the present research.

### 3.2. In vivo Toxicity Experiment of Mice

#### 3.2.1. Weight Changes of Mice

The weight changes of ICR mice for 5−10 weeks are shown in Table 5. Clearly, the body weight of the natural contaminated group (CC group) was lower than those of the three other groups, but the weight changes of the CC group was not significant (*p* > 0.05). This outcome might be due to the fact that ZEN and OTA concentrations were too low to cause weight changes. Mycotoxin-contaminated corn after EBI treatment (ECC group) was fed to mice and caused a slight growth of body weight compared with that of the CC group. However, the weight of the non-contaminated group after EBI treatment (ENC group) was lower than that of the non-contaminated group (NC group). This result might be due to the fact that irradiation can considerably influence corn quality, resulting in the poor growth conditions of mice.

For 103 weeks, Hooshmand et al. [35] fed B6C3F1 with fodders containing different ZEN concentrations, namely, 0, 8, and 17 mg/kg b.w./day for male mice and 0, 9, and 18 mg/kg b.w./day for female mice. All groups achieved similar survival rates and weight growths. Some researchers also fed FDRL Wistar mice with fodders containing 0, 25, and 50 mg/kg ZEN for 103 weeks and confirmed that the average weight growth of the experimental groups was lower compared with that of the control group. After 44 weeks, the average weights of male and female mice in the high-dose group decreased by 19% and 11%, respectively. Such reduction amplitude was related to the ZEN dose [36].

#### 3.2.2. Changes of the Organ Weights and Organ Indexes of Mice

Variations of the organ weights and organ indexes of the ICR mice at 10 weeks are shown in Table 6 and Table 7. Compared with the NC group, the CC group showed slightly higher liver, kidney, and spleen indexes but a lower testis index. However, no significant difference occurred between the two groups. The liver index of the ECC group was higher than those of other groups. In a word, all groups showed no significant differences in organ weights and organ indexes. Conkova et al. [37] fed FDRL Wistar rats with ZEN-containing fodders for 104 weeks (0, 0.1, 1, and 3 mg/kg b.w./day). The organ weights of male and female mice of the highest-dose group increased significantly. Compared with the above studies, the concentrations of contaminated fodders used in this experiment were relatively low and failed to cause significant changes to the organ weights and organ indexes of mice in the experiment. Moreover, EBI did not cause significant changes to the organ weights and organ indexes of ICR mice.

#### 3.2.3. Changes of Blood Indexes

The changes of the alanine transaminase (ALT), aspartate aminotransferase (AST), total bilirubin (TBIL), total protein (TP), albumin (ALB), blood urea nitrogen (BUN), and serum creatinine (SCr) of the different groups are shown in Table 8. The ALT, AST, TBIL, BUN, and SCr of the CC group were significantly higher than those of the NC group (*p* < 0.05), whereas the TP and ALB decreased dramatically (*p* < 0.05). These indexes of the ECC group were also improved considerably compared with those of the CC group, indicating that EBI can relieve the toxicity of mycotoxin to mice. The blood indexes of the ENC group were similar with those of the NC group. Thus, the fodder formula after EBI treatment can slightly influence the serum biochemical indexes of mice.

ZEN shows hepatotoxicity [38]. Gao et al. [39] revealed that the final weight of 17.9% mice decreased due to ZEN intake, thereby indicating that it induced liver damages. The increased activity of AST and alkaline phosphatase was enhanced, whereas the TP and ALB concentrations in the serum decreased, thereby causing histopathological lesions of the liver. Conkova et al. [37] investigated the influences of different concentrations (10 μg/kg b.w. and 100 μg/kg b.w.) of ZEN on the activities of AST, ALT, ALP, γ-GGT, and lactic dehydrogenase (LD) in rabbit serum. They verified that the ALP activity in the serum of the 10 μg dose group increased significantly after 168 h and 336 h of contamination. Moreover, AST, ALT, ALP, GGT, and LD activities in the serum of the 100 μg dose group after 168 h and 336 h contamination increased significantly, thereby indicating that the chronic effect of ZEN might cause hepatotoxicity. OTA shows nephrotoxicity [40]. OTA has been detected in serum samples of the same batch of pigs with OTA concentrations of 48.34−6.7 ng/mL to 84.2−41.17 ng/mL. These pigs all presented functional damages near the kidney tubules, and the urea nitrogen and creatinine concentrations in the blood increased [41]. In addition, ZEN can produce nephrotoxicity. Liang et al. found that after intraperitoneal injection of ZEN for three successive days under a single dose (50 mg/kg), mice suffered shrinkage of renal medullary substances and glomerulus, congestion of kidney tubules, swelling of the epithelial cells of the proximal convoluted tubule, and granular degeneration. After the intraperitoneal injection of ZEN at a single dose (100 mg/kg), urea and uric acid levels in mice increased significantly (*p* < 0.05) [42].

In this experiment, feeding with ZEN- and OTA-contaminated corn fodders induced functional damages to the liver and kidney of ICR mice. EBI treatment decreased the toxicity of the mycotoxin, thereby improving serum biochemical indexes and decreasing hepatotoxicity and nephrotoxicity.

#### 3.2.4. Liver and Kidney Histopathological Tissue Sections of Mice

ZEN and OTA may cause a series of histopathological liver and kidney changes. Therefore, observing the pathological changes of liver and kidney tissues of mice can be an effective way to evaluate EBI’s effects on the toxicity changes of a mycotoxin. In this study, the pathological liver and kidney changes of ICR mice were examined. Liver histopathological tissue sections of different groups are shown in Figure 2. Compared with those of the NC group, the liver tissues of the CC group presented slight nuclear heteromorphism and nuclear swelling. These phenomena were not observed in other groups. Liang et al. found fatty degeneration of hepatic cells and focal necrosis in the ZEN group. Few lymphocyte infiltrations were observed from some sections [42]. In renal histopathological tissue sections of ICR mice (Figure 3), the CC group developed no evident lesion compared with the NC group. Renal tissues of all groups remained relatively normal.

Hence, feeding with ZEN- and OTA-contaminated fodders can cause liver lesions in ICR mice but did not significantly influence the kidneys. To sum up, EBI treatment can effectively relieve the hepatotoxicity of mycotoxins.

## 4. Conclusions

The influences of EBI treatment on the physical and chemical properties of mycotoxin-contaminated corn were directly manifested by the reduced amylose content and starch crystallinity. Starch particles were broken, and starch appearance was destroyed along with the reduction of essential and total amino acid contents. Toxicity assessment revealed no significant differences among the ICR groups in terms of body weight, liver weight, and organ index. Blood indexes such as ALT, AST, TBIL, BUN, and SCr of mice fed with contaminated corn increased significantly (*p* < 0.05), whereas the TP and ALB decreased significantly (*p* < 0.05). Liver histopathological tissue lesions were observed, but no histopathological tissue lesion was found in the kidneys. Feeding mice with corn fodders after EBI treatment can improve blood biochemical indexes significantly. The resulting blood indexes were similar to those of normal mice, and no evident liver histopathological changes were noted. EBI treatment can influence some physical and chemical properties of contaminated corn, relieve the toxicity of mycotoxin in corn effectively, and increase the utilization of contaminated corn. This study provides theoretical reference and practical basis for the safety of EBI in treating mycotoxins in grains and the subsequent processing of grains in the future.

## Figures and Tables

**Figure 1 foods-09-00376-f001:**
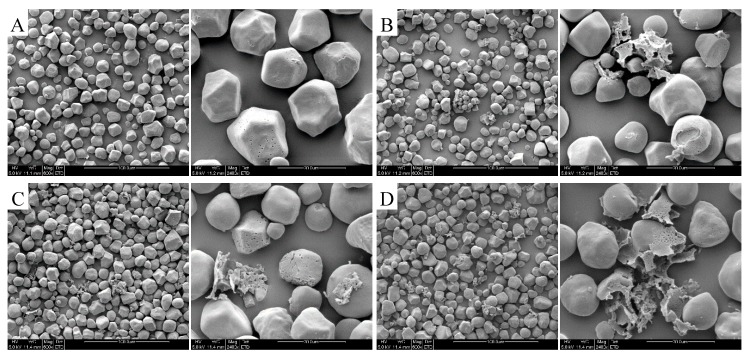
SEM of corn starch after EBI treatment (**A**) control, (**B**) 10 kGy, (**C**) 30 kGy, and (**D**) 50 kGy.

**Figure 2 foods-09-00376-f002:**
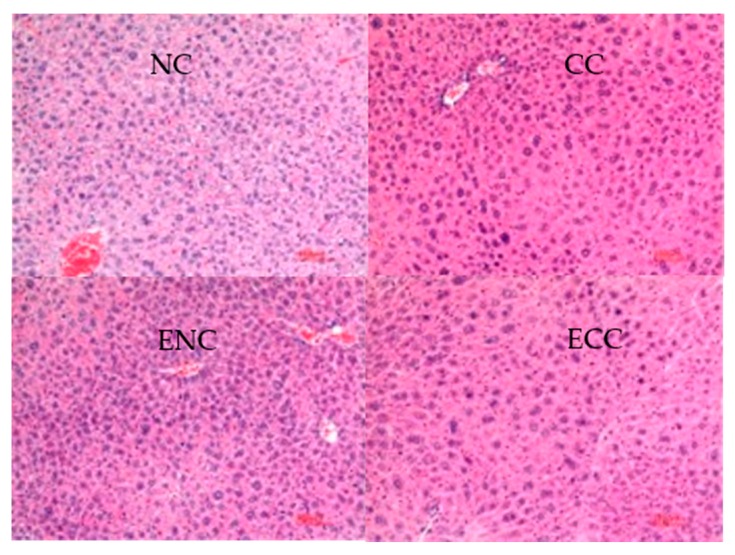
Liver histopathological tissue sections of Institute of Cancer Research (ICR) mice.

**Figure 3 foods-09-00376-f003:**
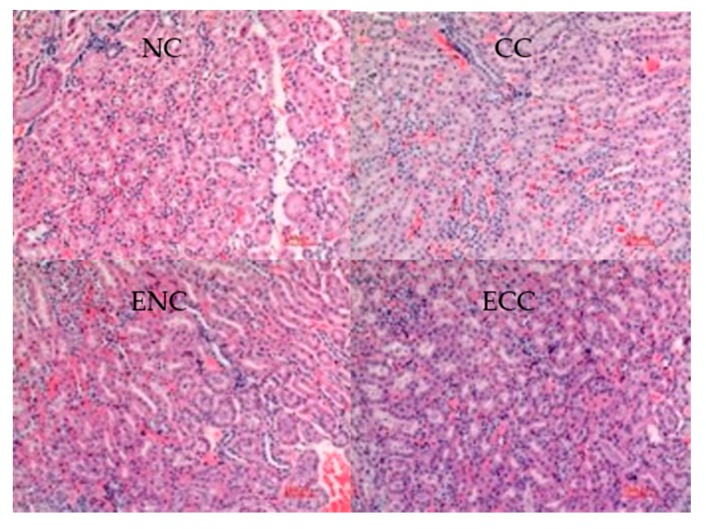
Kidney histopathological tissue sections of ICR mice.

**Table 1 foods-09-00376-t001:** Grouping of ICR mice.

Groups	Treatment Conditions	ZEN Content in Corn (μg/kg)	OTA Content in Corn (μg/kg)	ZEN Content in Fodders (μg/kg)	OTA Content in Fodders (μg/kg)
NC	Non-contaminated corn	-	-	-	-
CC	Contaminated corn	2875.79 ± 78.12	65.04 ± 8.39	1366.90 ± 39.06	30.89 ± 4.19
ENC	EBI treatment of non-contaminated corn	-	-	-	-
ECC	50 kGy EBI treatment of contaminated corn	815.62 ± 38.40	19.44 ± 4.65	346.64 ± 17.34	8.26 ± 0.35

Note: - reflects undetected values. NC: non-contaminated group. CC: natural contaminated group. ENC: non-contaminated group after electron beam irradiation (EBI) treatment. ECC: natural contaminated group after EBI treatment.

**Table 2 foods-09-00376-t002:** Corn amylase content after EBI treatment.

Doses (kGy)	5	10	20	30	50
Amylose (%)	22.28 ± 0.16 ^a^	23.28 ± 0.32 ^a^	16.86 ± 0.88 ^b^	14.11 ± 0.16 ^bc^	10.19 ± 0.88 ^d^

Note: Different letters in the same row represent significant differences (*p* < 0.05).

**Table 3 foods-09-00376-t003:** Starch crystallinity of corn after EBI treatment.

Sample	Control	10 kGy	30 kGy	50 kGy
Crystallinity (%)	10.9 ± 1.4 ^a^	10.0 ± 1.2 ^a^	8.2 ± 1.1 ^b^	7.6 ± 0.8 ^b^

Note: Sample at 0 kGy was used as control. Different letters in the same row represent significant differences (*p* < 0.05).

**Table 4 foods-09-00376-t004:** Corn amino acid content after EBI treatment.

Amino Acid (g/100 g)	Control	5 kGy	10 kGy	20 kGy	30 kGy	50 kGy
Asp	0.88 ± 0.13 ^a,b^	0.81 ± 0.07^a,b^	0.90 ± 0.05 ^a^	0.80 ± 0.03 ^b^	0.66 ± 0.02 ^c^	0.67 ± 0.01 ^c^
Glu	1.91 ± 0.16 ^a^	1.72 ± 0.12 ^a,b^	1.69 ± 0.15 ^a,b^	1.72 ± 0.15 ^a,b^	1.58 ± 0.11 ^b^	1.69 ± 0.13 ^a,b^
Ser	0.32 ± 0.01 ^a^	0.26 ± 0.01 ^b,c^	0.24 ± 0.02 ^c^	0.21 ± 0.01 ^c^	0.24 ± 0.02 ^c^	0.28 ± 0.01 ^b^
His	0.29 ± 0.01 ^a^	0.29 ± 0.01 ^a^	0.23 ± 0.02 ^b^	0.29 ± 0.02 ^a^	0.20 ± 0.02 ^b^	0.24 ± 0.01 ^b^
Gly	0.35 ± 0.02 ^b^	0.33 ± 0.02 ^b^	0.35 ± 0.04 ^a,b^	0.35 ± 0.01 ^b^	0.33 ± 0.01 ^b^	0.40 ± 0.02 ^a^
Thr	0.31 ± 0.01 ^a,b^	0.32 ± 0.01 ^a^	0.27 ± 0.03 ^b^	0.22 ± 0.01 ^c^	0.23 ± 0.01 ^b,c^	0.17 ± 0.02 ^d^
Arg	0.96 ± 0.12 ^a^	0.94 ± 0.03 ^a^	0.87 ± 0.04 ^a^	0.84 ± 0.03 ^a,b^	0.79 ± 0.02 ^b^	0.85 ± 0.03 ^a^
Ala	0.62 ± 0.11 ^a,b^	0.70 ± 0.03 ^a^	0.60 ± 0.03 ^b^	0.59 ± 0.02 ^b^	0.56 ± 0.01 ^b^	0.54 ± 0.04 ^b^
Tyr	0.37 ± 0.02 ^a^	0.36 ± 0.03 ^a,b^	0.34 ± 0.01 ^a,b^	0.35 ± 0.03 ^a,b^	0.32 ± 0.01 ^b^	0.32 ± 0.01 ^b^
Cys	0.16 ± 0.04 ^a^	0.14 ± 0.03 ^a^	0.16 ± 0.04 ^a^	0.10 ± 0.02 ^a^	0.13 ± 0.02 ^a^	0.13 ± 0.04 ^a^
Val	0.45 ± 0.02 ^b^	0.33 ± 0.03 ^c^	0.32 ± 0.01 ^c^	0.34 ± 0.01 ^c^	0.45 ± 0.03 ^b^	0.51 ± 0.02 ^a^
Met	0.16 ± 0.01 ^a^	0.11 ± 0.02 ^b^	0.17 ± 0.01 ^a^	0.14 ± 0.02 ^a,b^	0.12 ± 0.02 ^b^	0.11 ± 0.01 ^b^
Phe	0.41 ± 0.03 ^a^	0.45 ± 0.03 ^a^	0.40 ± 0.02 ^a^	0.33 ± 0.01 ^b^	0.30 ± 0.02 ^b,c^	0.29 ± 0.02 ^c^
Ile	0.33 ± 0.02 ^a^	0.25 ± 0.01 ^b^	0.24 ± 0.01 ^b^	0.23 ± 0.02 ^b^	0.25 ± 0.04 ^b^	0.30 ± 0.02 ^a,b^
Leu	0.96 ± 0.07 ^a^	0.72 ± 0.03 ^b^	0.70 ± 0.02 ^b^	0.70 ± 0.02 ^b^	0.63 ± 0.02 ^c^	0.64 ± 0.05 ^b,c^
Lys	0.32 ± 0.04 ^a^	0.28 ± 0.01 ^a^	0.21 ± 0.02 ^b^	0.22 ± 0.02 ^b^	0.23 ± 0.02 ^b^	0.14 ± 0.02 ^c^
Pro	0.53 ± 0.05 ^a^	0.48 ± 0.01 ^a^	0.46 ± 0.03 ^a^	0.47 ± 0.03 ^a^	0.40 ± 0.02 ^b^	0.37 ± 0.01 ^b^
EAA	2.92 ± 0.06 ^a^	2.46 ± 0.09 ^b^	2.31 ± 0.04 ^c^	2.19 ± 0.05 ^d^	2.20 ± 0.05 ^d^	2.16 ± 0.01 ^d^
TAA	9.31 ± 0.08 ^a^	8.49 ± 0.07 ^b^	8.15 ± 0.05 ^c^	7.89 ± 0.09 ^d^	7.42 ± 0.02 ^f^	7.64 ± 0.02 ^e^
EAA/TAA	0.31 ± 0.04 ^a^	0.29 ± 0.05 ^a^	0.28 ± 0.07 ^a^	0.28 ± 0.10 ^a^	0.30 ± 0.01 ^a^	0.28 ± 0.08 ^a^

Note: Sample at 0 kGy was used as control. Different letters in the same row represent significant differences (*p* < 0.05), whereas the same letters in the same row represent insignificant differences (*p* > 0.05).

**Table 5 foods-09-00376-t005:** Body weights of ICR mice.

Weekly Age	5 (g)	6 (g)	7 (g)	8 (g)	9 (g)	10 (g)
NC	28.5 ± 2.7 ^a^	30.4 ± 2.6 ^a^	32.3 ± 2.7 ^a^	34.3 ± 2.6 ^a^	36.2 ± 2.6 ^a^	38.2 ± 2.5 ^a^
CC	28.4 ± 2.6 ^a^	30.1 ± 2.5 ^a^	31.7 ± 2.4 ^a^	33.2 ± 2.5 ^a^	34.5 ± 2.2 ^a^	36.0 ± 2.2 ^a^
ENC	29.4 ± 2.0 ^a^	31.0 ± 2.0 ^a^	32.6 ± 1.9 ^a^	34.1 ± 1.9 ^a^	35.6 ± 1.9 ^a^	37.3 ± 1.8 ^a^
ECC	29.5 ± 1.8 ^a^	31.1 ± 1.8 ^a^	32.6 ± 1.8 ^a^	34.2 ± 1.7 ^a^	35.7 ± 1.7 ^a^	37.4 ± 1.7 ^a^

Note: NC: non-contaminated group. CC: natural contaminated group. ENC: non-contaminated group after EBI treatment. ECC: natural contaminated group after EBI treatment. Same letters in the same column represent insignificant differences (*p* > 0.05).

**Table 6 foods-09-00376-t006:** Organ weights of ICR mice.

Groups	Weight of Liver (g)	Weight of Kidney (g)	Weight of Spleen (g)	Weight of Testis (g)
NC	1.65 ± 0.18 ^a^	0.61 ± 0.05 ^a^	0.10 ± 0.02 ^a^	0.30 ± 0.03 ^a^
CC	1.73 ± 0.24 ^a^	0.60 ± 0.06 ^a^	0.11 ± 0.03 ^a^	0.26 ± 0.04 ^a^
ENC	1.68 ± 0.19 ^a^	0.64 ± 0.08 ^a^	0.12 ± 0.02 ^a^	0.30 ± 0.04 ^a^
ECC	1.80 ± 0.17 ^a^	0.67 ± 0.07 ^a^	0.13 ± 0.02 ^a^	0.29 ± 0.05 ^a^

Note: NC: non-contaminated group. CC: natural contaminated group. ENC: non-contaminated group after EBI treatment. ECC: natural contaminated group after EBI treatment. Same letters in the same column represent insignificant differences (*p* > 0.05).

**Table 7 foods-09-00376-t007:** Organ indexes of ICR mice.

Groups	Liver Index (%)	Kidney Index (%)	Spleen Index (%)	Testis Index (%)
NC	4.33 ± 0.37 ^a^	1.60 ± 0.12 ^a^	0.27 ± 0.04 ^a^	0.80 ± 0.10 ^a^
CC	4.95 ± 0.66 ^a^	1.74 ± 0.20 ^a^	0.33 ± 0.09 ^a^	0.74 ± 0.09 ^a^
ENC	4.51 ± 0.49 ^a^	1.72 ± 0.18 ^a^	0.33 ± 0.06 ^a^	0.80 ± 0.14 ^a^
ECC	4.83 ± 0.44 ^a^	1.80 ± 0.19 ^a^	0.36 ± 0.07 ^a^	0.77 ± 0.14 ^a^

Note: NC: non-contaminated group. CC: natural contaminated group. ENC: non-contaminated group after EBI treatment. ECC: natural contaminated group after EBI treatment. Same letters in the same column represent insignificant differences (*p* > 0.05).

**Table 8 foods-09-00376-t008:** Blood indexes of ICR mice.

-	Indexes						
Groups	ALT	AST	TBIL	TP	ALB	BUN	SCr
-	(IU/L)	(IU/L)	(μmol/L)	(g/L)	(g/L)	(mmol/L)	(μmol/L)
NC	22 ± 2 ^c^	86 ± 5 ^c^	2.88 ± 0.40 ^b^	62.6 ± 3.5 ^a^	38.0 ± 2.8 ^a^	7.6 ± 0.3 ^b^	14.4 ± 0.8 ^b^
CC	66 ± 15 ^a^	172 ± 32 ^a^	4.76 ± 0.46 ^a^	46.0 ± 4.8 ^c^	23.2 ± 6.3 ^c^	9.4 ± 1.2 ^a^	20.4 ± 2.5 ^a^
ENC	19 ± 4 ^c^	110 ± 5 ^b^	2.56 ± 0.35 ^b^	56.0 ± 0.8 ^b^	33.0 ± 1.8 ^b^	7.1 ± 0.2 ^b,c^	14.8 ± 0.6 ^b^
ECC	30 ± 3 ^b^	134 ± 18 ^a^	2.56 ± 0.26 ^b^	59.9 ± 2.8 ^a^	36.1 ± 1.5 ^a,b^	6.8 ± 0.1 ^c^	15.7 ± 0.8 ^b^

Note: ALT: alanine transaminase. AST: aspartate aminotransferase. TBIL: total bilirubin. TP: total protein. ALB: albumin. BUN: blood urea nitrogen. SCr: serum creatinine. Different letters in the same column represent significant differences (*p* < 0.05), whereas the same letters in the same column represent insignificant differences (*p* > 0.05).

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
