# Peer review of "Influences of Electron Beam Irradiation on the Physical and Chemical Properties of Zearalenone- and Ochratoxin A-Contaminated Corn and In Vivo Toxicity Assessment"

_foods, 2020, doi:10.3390/foods9030376_

Round 1
Reviewer 1 Report
In the title authors name a mycotoxin however in the abstract they include two different mycotoxins
In the Abstract the following sentence should be clarify: “The EBI-treated group was not significantly different from the normal corn group and displayed histopathological changes of the liver”; Authors say that the groups treated did not show significant differences, however they say I the same sentence that the they displayed histopathological changes.
In the materials and method section where it is said that the columns and detector were made by Agilent it should be said that they were manufactured; or that they were provide by Agilent.
Author should explain in more extent how they contaminate the corn, and what concentration of mycotoxins they used.
They also should be include the methodology and analysis done to this mycotoxins in corn.
In section 3.1.1 there is contradictory information referred to the effect of high irradiation dose in the concentration of amylose.
Authors also should provide data and discuss more about the effect of the different radiations on the concentration of mycotoxins in corn.
Authors should discuss and relate the changes in corn structure produced by the irradiation and their effect on mycotoxin regarding their concentrations.
Statistical SPSS tests should be applied to evaluate the differences (significant or not) on amino acid composition in Table 4. The same for Tables 5 and 6, related with the effect of irradiation on body and organ weights of mice.
Additionally, in Tables 7 and 8 also should be included the statistical treatment of data.
In case that does not exist significant different between samples, a foot note should be included in the Table, and in the case that exist significant differences between samples, different letters should be included for different groups.
Reviewer 2 Report
The manuscript presents interesting results concerning appropriate application of Electron Beam Radiation for reducing the concentration of ZEN and OTA in corn samples taking into account the results of in vitro and in vivo tests. These results might be useful for practical application. However, this manuscript requires improving due to some shortcomings.
Comments that should be considered to make the manuscript suitable for publication:
L68. “Tthe moisture content and sugar reduction increased significantly” – this statement can be misleading because it can be understood that there is an intensification of reduction for both the water content and sugar content, or that the water content increases and the sugar content decreases. Use a more explicit wording.
L81. Consider replacing “corn” by “corn kernels”. The concentration of ZEN and OTA in collected raw material is missing.
L97-101. Provide details (type, manufacturer) for scanning electron microscope and X-ray diffractometer.
L103. Explain in which way the two levels of moisture content were obtained. Why these two levels of moisture content were considered? What was the method of moisture content determination?
L105. The same weight of samples is crucial to determine the unit dose of radiation. After sieving through a 30 mesh screen, the weight of the milled sample was probably reduced. It is not clear how much.
L106. “After” or perhaps “before” EBI treatment?
L130-131. Table 1. It was stated before in Line 81 that contaminated corn kernels were collected from farmer’s markets. Please explain in which way contaminated and non-contaminated samples were prepared? This table concerns rather corn samples than IRC mice and should be presented in the Results and Discussion section.
L157-229. It is not clear whether the results concern whole or ground kernels. The preparation of corn samples was described in Lines 102-108.
L331. It is hard to say „contaminated mice” as mice was fed with contaminated corn samples.
Reviewer 3 Report
- Potential application of developed method should be mentioned in Abstract.
- Elements of scientific novelty should be presented in a detailed and convincing manner (in the last paragraph of the Introduction, and shortly in Abstract).
- I suggest that a diagram (scheme) presenting the developed analytical procedure used in the study should be extended as in this form is not easy readable. It would help understand the details of the analytical protocol better, and allow the written description of the procedure to be shortened.
- Application of proper quality assurance/quality control (QA/QC) procedures is vital for the
measurement results to be treated as a source of reliable analytical information. Consequently, I suggest that a separate section devoted to QA/QC be added to the manuscript. Special attention should be paid to:
- description of the validation procedure for the applied/proposed analytical protocol,
- information on metrological characteristics of the analytical procedure, especially Method
Quantitation Limit (MQL) values for the entire procedure (from handling of representative samples to statistical and chemometric evaluation of the data sets obtained), and not only for the analytical techniques used during the analysis of the extracts.
- Because developed procedure seems to be green taking into account the principles of Green Analytical Chemistry, it will be perfect to present also this kind of results, and maybe green aspects of different approaches known from the literature should be also shortly discussed. Please consider such tools: the Analytical Eco-Scale [1] and Green Analytical Procedure Index [2] to compare the results.
[1] A. GaÅ‚uszka, Z.M. Migaszewski, P. Konieczka, J. NamieÅ›nik, Analytical Eco-Scale for assessing the greenness of analytical procedures. Trends Anal. Chem., 2012, 37, 61–72.
[2] J. PÅ‚otka-Wasylka, A new tool for the evaluation of the analytical procedure: Green Analytical Procedure Index, Talanta, 2018 In press DOI: https://doi.org/10.1016/j.talanta.2018.01.013
- Authors should compare the developed procedure with existing methodologies taking into account green character of these procedures but also chromatographic and validation parameters
- 7. Innovative potential of the results obtained should be explained in detail (CONCLUSIONS).
- Please correct the amounts with the corresponding errors (value ± SD). It is written in wrong way.Please correct it in the tabels and text.
Round 2
Reviewer 1 Report
Authors have realized all the changes proposed in the previous revision, thus te manuscript can be accepted in the present form
Reviewer 3 Report
I do not agree with the some point but respect your opinion, thus, accept this version